# Land-Cover Dependent Relationships between Fire and Soil Moisture

**Alexander J. Schaefer [1,2] and Brian I. Magi [2,\*]** 

[1] Program in Infrastructure and Environmental Systems, The William States Lee College of Engineering, Charlotte, University of North Carolina at Charlotte, Charlotte, NC 28223, USA; aschae11@uncc.edu

[2] Department of Geography and Earth Sciences, University of North Carolina at Charlotte, Charlotte, NC 28223, USA

\* Correspondence: brian.magi@uncc.edu

**Abstract:** For this study, we characterized the dependence of fire counts (FCs) on soil moisture (SM) at global and sub-global scales using 15 years of remote sensing data. We argue that this mathematical relationship serves as an effective way to predict fire because it is a proxy for the semi-quantitative fire–productivity relationship that describes the tradeoff between fuel availability and climate as constraints on fire activity. We partitioned the globe into land-use and land-cover (LULC) categories of forest, grass, cropland, and pasture to investigate how the fire–soil moisture (fire–SM) behavior varies as a function of LULC. We also partitioned the globe into four broadly defined biomes (Boreal, Grassland-Savanna, Temperate, and Tropical) to study the dependence of fire–SM behavior on LULC across those biomes. The forest and grass LULC fire–SM curves are qualitatively similar to the fire–productivity relationship with a peak in fire activity at intermediate SM, a steep decline in fire activity at low SM (productivity constraint), and gradual decline as SM increases (climate constraint), but our analysis highlights how forests and grasses differ across biomes as well. Pasture and cropland LULC are a distinctly human use of the landscape, and fires detected on those LULC types include intentional fires. Cropland fire–SM curves are similar to those for grass LULC, but pasture fires are evident at higher SM values than other LULC. This suggests a departure from the expected climate constraint when burning is happening at non-optimal flammability conditions. Using over a decade of remote sensing data, our results show that quantifying fires relative to a single physical climate variable (soil moisture) is possible on both cultivated and uncultivated landscapes. Linking fire to observable soil moisture conditions for different land-cover types has important applications in fire management and fire modeling.

**Keywords:** soil moisture; remote sensing; fire counts; land use

## 1. Introduction

Present day fires burn 350–450 million hectares globally per year [1,2], and release about 2.5 petagrams of carbon into the atmosphere annually [3]. Understanding the relationship among fire, ecosystem, and human activities is crucial to advancing our capacity to predict fire danger and occurrence. Fire models strive to simulate the spatial and temporal distribution of global fire by modeling the dependence of fire occurrence and spread on the underlying environmental variables [4–6], while observationally based data from satellites, paleoproxies, and historical records inform how we understand interactions among fire, climate, and human activity and help to refine approaches within fire models [7,8].

One major challenge to modeling, observational, and paleoproxy approaches in fire science is that the importance of the drivers of global fire activity varies as a function of the spatial and



temporal scales being considered [9–11]. At the largest scales, climate is the controlling factor of fire activity [12–14], where fire occurrence is linked to variability in seasonal soil moisture (SM) (e.g., [4,15–20]), precipitation [21], temperature (e.g., [4,14,22–24]), and net primary production (NPP) [4,22,24,25]. To varying degrees, these environmental factors determine the combustibility and fuel load of any landscape. In the classical approach to fire science, an ignition source either in the form of lightning or a human (combined with the environmental factors) is required to generate a fire start.

The balance of climate variability with ecosystem productivity in determining the amount of fire activity can be described using a semi-quantitative conceptual framework that has been referred to as the fire–productivity relationship [26]. The relationship suggests that fuel loads increase as NPP increases, but that there is a tradeoff because the combustibility of the vegetation decreases due to the increased moisture [22,26]. On the other hand, fuel loads decrease as NPP decreases but these ecosystems with lower NPP tend towards climatic conditions that are more flammable even as they become fuel-limited [22,25,27]. In the fire–productivity conceptual model then, the productivity and climate constraints surround a zone of optimal fire conditions. Studies have confirmed this productivity tradeoff with climatological combustibility [4,22,26,27], but they have focused on studying this behavior in biomes and not for specific land-cover types within those biomes.

Human influences on fire activity are evident in present-day observations of fire both in terms of clear shifts in fire seasonality [28,29] as well as through associations of documented human use of fire at the landscape scale and satellite fire records [30], all of which emphasizes the need to consider human agency in fire ecology. Linking human fire use to population density alone is problematic [4,30], but neglecting human influences altogether in causal or predictive studies can overemphasize the relationship between fire activity and climatic variables [5,31], with results that have systematic and time-varying inaccuracies over both short (present day) and long (paleo, or future) time horizons [28,32–37]. For example, pasture is both a large fraction of global land area and global burned area [37], and increasing pasture fraction corresponds to an increase in the probability of burning [4]. A question that remains to be tested is whether the fire–productivity relationship holds on cultivated landscapes such as pasture and cropland.

This study leverages 15 years of satellite observations and retrievals of fire counts (FCs) and soil moisture to evaluate how fire counts relate to soil moisture for four major land-use and land-cover (LULC) types that include grass, forest, cropland, and pasture at the global scale. We suggest that the fire–soil moisture (fire–SM) relationships we derive are analogous to the fire–productivity relationship [22,25,26], and evaluate how the fire–SM relationship varies across LULC in large sub-regions of the globe. We also evaluate how these variations reflect ecophysiological characteristics of the dominant plants and human influences. The main objective of this study was to test the hypothesis of varying constraints on the fire–productivity relationship [22] using our fire–SM curves. Our study helps lay the groundwork for a new approach for predicting fire counts using concurrent soil moisture conditions not only in semi-natural grasses and forests, but also on the more challenging to characterize cropland and pasture land cover.

## 2. Materials and Methods

### 2.1. Soil Moisture Data

We produced a monthly soil moisture (SM) product using the European Space Agency (ESA) Climate Change Initiative (CCI) combined SM product version 4.2, which estimates SM in the top 5 cm of soil by fusing daily active and passive satellite radiometer swaths into a unified gridded product [38–42]. The ESA CCI combined product is distributed on a quarter-degree grid and spans from November 1978 through December 2016, although more reliable global coverage begins in the early 2000s. We averaged the daily ESA CCI data into a time series of monthly mean values from 2000 to 2014 (Figure 1 shows the monthly values averaged over all years). The number of daily data points in any given month varies (Figure S1, Supplementary Materials) because satellite soil moisture retrievals

are not possible for areas of dense vegetation (e.g., tropical forests) or sub-zero soil temperatures. The amount of data contributing to a particular month affects whether a monthly average is calculated using only a few days or a full month of days. Overall, Figure 1 is similar to Forkel et al. [43] except we did not attempt to fill in temporal gaps in the soil moisture time series with a mathematical regression model. Thus, our monthly soil moisture time series has gaps in regions such as northern latitudes due to seasonal snow cover.

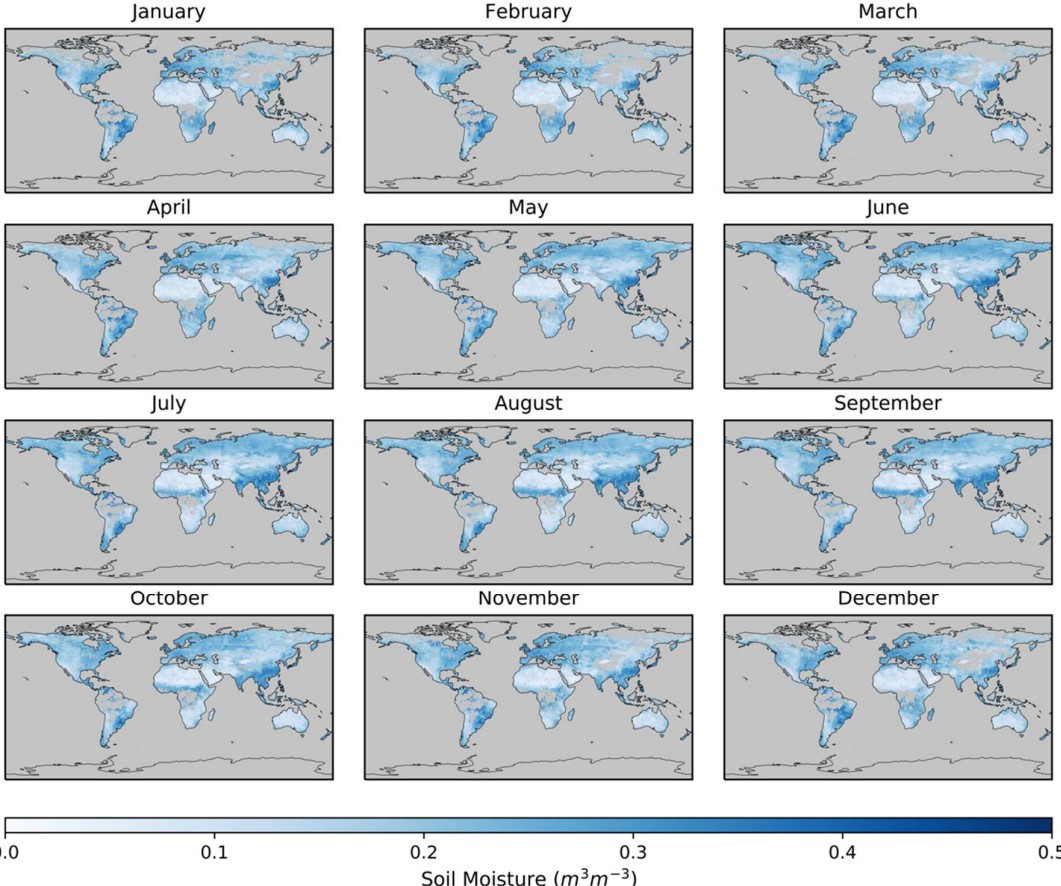

**Figure 1.** Mean monthly soil moisture calculated from European Space Agency (ESA) Climate Change Initiative (CCI) Soil Moisture 4.2 for 2000 through 2014. Grey denotes areas with no data.

### 2.2. Fire Count Data

We produced fire count (FC) maps using NASA Moderate Resolution Imaging Spectrometer (MODIS) Collection 6 [44,45] data. The algorithm uses thermal infrared emissions to detect and locate fires as small as 50 m$^2$ under ideal conditions, with an average fire detection size of about 900 m$^2$ [46]. We aggregated the data as the total number of fire counts per month within each quarter-degree grid cell. MODIS data are available through the present, but we used the period 2000–2014 to match other datasets in our analysis.

### 2.3. Land Cover

The Land-Use Harmonization version 2 (LUH2) [47–51] dataset is a historical reconstruction of land use from 850 to 2015 with future projections that extend to 2100 and is distributed on a quarter-degree grid. We used LUH2 annual land-use and land-cover (LULC) fractions from 2000 to 2014. LUH2 includes many LULC sub-categories [49] that we combined to create four categories—grass, forest, pasture, cropland—for our analysis as follows. Grass land cover is a combination of primary and secondary grassland; forest includes both primary and secondary forest; pasture is composed of

rangeland and managed pasture; and cropland is a combination of five crop types ranging from annual to perennial to nitrogen-fixing crops. There are other land-cover datasets [52], but the primary reason we used LUH2 is the explicit designation of the fraction of land covered in pasture.

### 2.4. Biomes

The global distribution of biomes is from Levavasseur et al. [53] who used a multinomial logistic model to objectively map the first and second most probable biomes on a one-sixth-degree grid. We remapped the biome data to a quarter-degree grid (the grid size for our analysis) using nearest-neighbor interpolation and aggregated the original biomes into four broadly defined biomes as follows: (1) Boreal (BORL) which combines boreal forest and tundra; (2) Grassland-Savanna (GRSA) combines grassland, dry shrubland, desert vegetation, and warm desert; (3) Temperate (TEMP) combines temperate forest and warm-temperate forest when the second most probable biome is not tropical forest; and (4) Tropical (TROP) combines tropical forest with warm-temperate forest when the second most probable biome for warm-temperate forest is tropical forest. Each biome contains a varying fraction of different LULC types, as specified by LUH2 (Section 2.3). The biome data are static over the timespan of our analysis (2000–2014).

### 2.5. Data Analysis

Soil moisture (SM) and fire count (FC) datasets were analyzed in the four LULC categories (forest, grass, cropland, and pasture), but since LULC is defined as a fraction of the grid cell, we only analyzed grid cells with LULC greater than a threshold value. In grid cells that met the threshold requirement, we assumed all the FCs were associated with that particular LULC type. We tested LULC thresholds ranging from 50% to 100%, and selected 75% to balance the following two points: (1) maximize the confidence that the FCs in the grid cell are actually associated with that LULC type and (2) preserve enough data for meaningful results.

Figure 2 shows the global distribution of the four LULC categories with cells having greater than 75% of that particular LULC type colored in red. It also shows that the only LULC that does not have vast global coverage at 75% is cropland, and we discuss this lack of spatial representativeness below (Section 3.3). Our analysis of the fire–SM relationship is based on the grid cells in red.

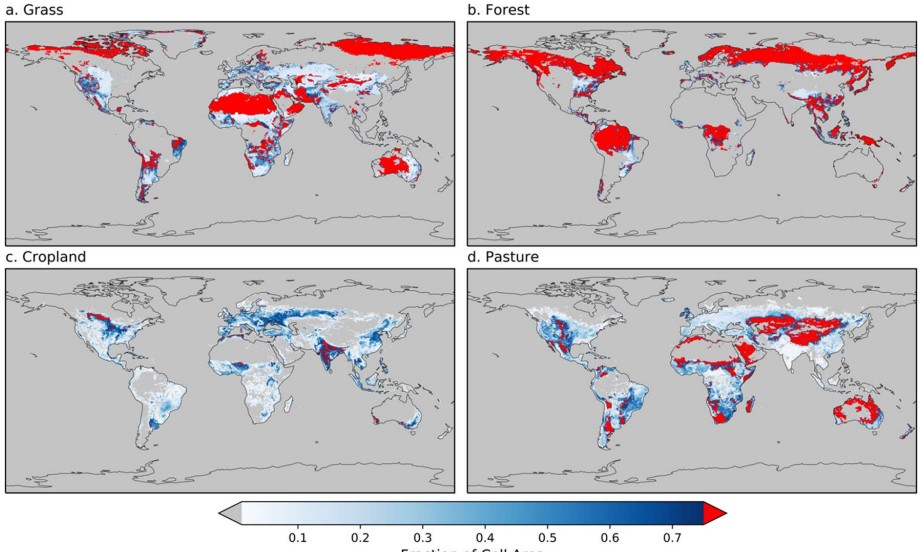

**Figure 2.** The global distribution of land-use and land-cover (LULC) types: (**a**) grass, (**b**) forest, (**c**) cropland, and (**d**) pasture. Red indicates grid cells with greater than 75% of that particular LULC type, and grey denotes that the grid cell has zero of that LULC type.

After applying the 75% LULC threshold, the remaining SM and FC data were extracted and average FCs were calculated for SM bins with a width of 0.05 m$^3$/m$^3$, noting that fire counts of zero were included. For displaying results on the same vertical scale, average FC values were normalized to the maximum average FC.

## 3. Results and Discussion

Figure 3 shows average FCs (normalized to the maximum average FC listed in Table S1, Supplementary Materials) as a function of SM for the global domain and as a function of LULC type. Analogous figures showing burned area are included in the Supplementary Materials (Figure S2, Supplementary Materials), but these are similar to fire–SM curves using fire counts, likely since the burned area dataset is based heavily on MODIS fire counts [3,54]. FCs vary widely in each SM bin (Figure S3, Supplementary Materials).

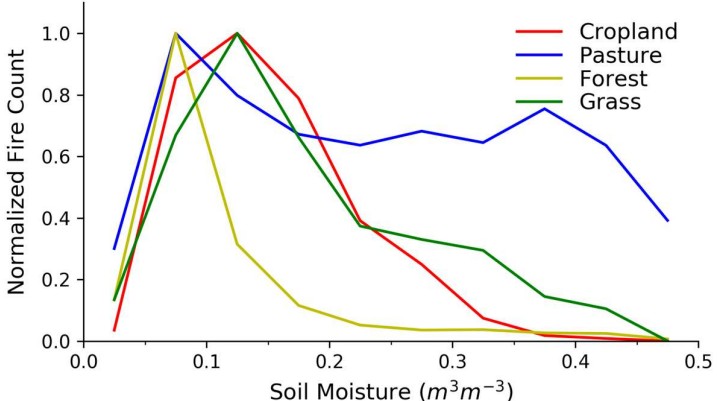

**Figure 3.** Relationships between fire counts (FCs) and soil moisture (SM) across different land-use and land-cover types. Fire counts are normalized to the maximum average FC in each LULC category with normalization constants in Table S1.

The fire–SM relationships in Figure 3 vary across LULC categories, but they each peak in FCs at low SM between about 0.05–0.15 m$^3$/m$^3$, sharply falling off at lower SM values while decreasing slowly as SM increases. This general behavior is qualitatively similar to the primary characteristic of the fire–productivity relationship [22,26]: Fire activity tends towards a maximum value at an intermediate value of ecosystem productivity and aridity. This similarity is expected since higher SM values are generally related to higher fuel production, even while the combustibility of those fuels decreases. We argue that the results in Figure 3 are analogous to the productivity and climate constraints where the fire–SM curves generally decrease when moving away from intermediate SM values. We discuss how this general result varies across LULC types and also across biomes.

Compared with grass, cropland, and pasture land cover, fires in forested land cover decrease more rapidly as SM increases, whereas the fire–SM relation for pasture suggests a relatively high fire danger at relatively large SM values (Figure 3). Grass and pasture share overlapping physical characteristics [49,55], but Figure 3 shows divergence at SM values greater than ≈0.2 m$^3$/m$^3$, suggesting that while the global pasture fire–SM may be similar to the fire–SM curve for global grasses at lower SM, there is a greater risk of pasture fires at higher SM values than for grasses. Our result is consistent with the analysis performed by Bistinas et al. [4] indicating the increase in the probability of burning as pasture fraction increases.

To understand whether the global fire–SM relationships in Figure 3 are representative of behavior at sub-global scales, we partitioned our analysis into broadly defined biomes (see Section 2.4). Table 1 shows the percent of LULC in each biome both before and after the 75% threshold is applied, and it is clear that grass, cropland, and pasture LULC are largely located within the Grassland-Savanna biome, whereas forest LULC is in the Boreal, Tropical, and Temperate biomes. After the threshold is applied,

84% of cropland LULC in our analysis becomes concentrated in the Grassland-Savanna biome, but we discuss this in Section 3.3. Figure S4 (Supplementary Materials) shows LULC distributions across biomes, Figure S5 (Supplementary Materials) shows the FCs in each of these biomes, and Table S2 (Supplementary Materials) lists the number of fire counts after the threshold is applied.

**Table 1.** The fractional distribution of grass, forest, cropland, and pasture land use and land cover (LULC) across the biomes (Boreal BORL, Grassland-Savanna GRSA, Temperate TEMP, and Tropical TROP), with values after the 75% threshold is applied shown in round brackets.

|  | BORL BIOME (%) | GRSA BIOME (%) | TEMP BIOME (%) | TROP BIOME (%) |
|---|---|---|---|---|
| GRASS | 19.2 (25.5) | 67.6 (67.7) | 10.7 (5.4) | 2.5 (1.5) |
| FOREST | 41.4 (47.8) | 10.6 (7.8) | 22.0 (17.6) | 26.0 (26.9) |
| CROPLAND | 5.2 (4.4) | 50.7 (84.1) | 34.8 (11.2) | 9.3 (0.3) |
| PASTURE | 6.2 (7.9) | 71.2 (84.4) | 15.5 (4.8) | 7.1 (2.9) |

Figure 4 shows how the fire–SM relationship for a particular LULC type varies as a function of biome relative to the global fire–SM relationship. The fire–SM relationship across LULC types within a biome is given in Figure S6 (Supplementary Materials), and similar figures for burned area are included in Figures S7–S9 (Supplementary Materials). Normalization factors for Figure 4 are found in Table S1. We discuss the results of each land-cover type shown in Figure 4 in the sections below.

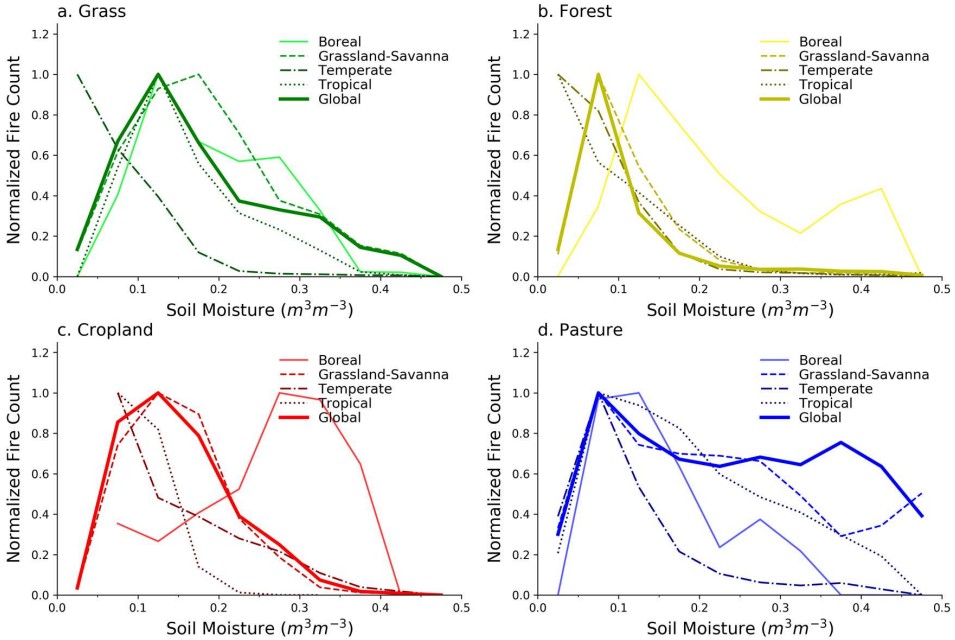

**Figure 4.** The relationships between fire counts and soil moisture across biomes for specific LULC types: (**a**) grass, (**b**) forest, (**c**) cropland, and (**d**) pasture. "Global" represents the fire–SM curves shown in Figure 3. All fire counts are normalized to the maximum average value for each LULC, with normalization constants in Table S1.

## 3.1. Grass Land Cover

The fire–SM relationship for grass LULC (Figure 4a) in the Boreal, Grassland-Savanna, and Tropical biomes is similar for low to intermediate SM, and tends towards the grass curve for Grassland-Savanna at higher SM. Grass in the Temperate biome exhibits a different dependence (Figure 4a), but only 5.4% of grass is in the Temperate biome after the threshold is applied (Table 1). A closer examination of the spatial locations of these grasslands (Figure S4) and the fires in those locations (Figure S5) shows that much of that unique Temperate grass fire–SM behavior is based on data from southern Africa, especially southeast of the Democratic Republic of Congo. This area is located alongside a landscape

that is woody savanna, and the grass fire–SM curve is similar to the curve for forests in the Temperate biome (Figure 4b). Furthermore, as discussed by Levavasseur et al. [53], this particular region of southern Africa has a high uncertainty in its biome designation, marking it as a borderline climate that could potentially support warm temperature forest or grassland. In general, grass LULC in the Grassland-Savanna and Boreal biomes (where over 90% of grass is located) exhibits peak fire counts at low-intermediate SM values, and a falloff in fire danger away from the peak. The analog in fire–productivity is that low SM values correspond to a low-productivity landscape (fuel constrained), and higher SM values reflect grasses that are less combustible fuels (climate constrained).

## 3.2. Forest Land Cover

The fire–SM relationship in forests is generally similar across biomes (Figure 4b), but in the Boreal biome, which contains 48% of forests analyzed (Table 1), the fire–SM curve departs from others and peaks at higher SM with a long tail that indicates fires occur often even at those higher SM values (relative to forests in other biomes). This behavior may reflect less dependence on synchronous SM in Boreal forests since intermediate-high levels of SM ranging from 0.2 to 0.4 $m^3/m^3$ do not seem to deter fires in the Boreal forests. Other biomes, however, show that the maximum amount of fire tends to occur at a fairly narrow range of SM values of about 0.05–0.125 $m^3/m^3$. This narrower range of fire danger may be a result of higher resistance to fire occurrence in those forests due to the deeper root systems that allow access to water well below the surface [56], recalling that the satellite-based SM in our study tends to reflect near-surface SM conditions. Boreal forests are more prone to fire over a far wider range of SM values, and exhibit increased sensitivity when SM is particularly low, which might arise during drought conditions linked to persistently higher temperatures.

While there is a clear falloff in the Boreal and Grassland-Savanna forests as SM decreases, the forests in the Temperate and Tropical biomes, which account for 45% of forest in our analysis (Table 1), show no falloff in fire at low SM. Climatologically low SM usually would be expected to coincide with lower ecosystem productivity and fuel availability (and less fire), but perhaps Figure 4b suggests that even when Temperate and Tropical forests dry out, the short-term ecosystem response is still one that supports fire because the fuels remain in place even when near-surface SM is low.

Similar to grasses, forest fire–SM curves reflect the productivity and climate constraints in fire–productivity space, but forests generally have a narrower range of SM values that produce optimal fire danger conditions. The exception is the Boreal forests with a fire–SM curve that is more similar to fire–SM curves for grasses than other forests, which is perhaps a result of the shallower mean root depth of the dominant tree species in Boreal forests being quantitatively similar to the shallow root depths of grass species [56]. We suggest that near-surface SM therefore captures productivity and climate constraints for forests, but in ways that depend on biome, and are generally different than grasses except where trees tend to have shallow rooting systems.

## 3.3. Cropland Land Cover

The Grassland-Savanna biome contains 84% of analyzed cropland, while most of the remaining cropland is in the Temperate biome (Table 1), so the global curve closely resembles the Grassland-Savanna curve (Figure 4c). After the 75% threshold is applied in the Tropical biome, only 0.3% of cropland remains in the analysis (concentrated in a tiny section of West Africa), so the fire–SM relationship in that biome is unlikely to be representative of cropland more broadly in that biome. For Boreal cropland, the average FC increases as SM increases suggesting a very different fire–SM relationship than cropland in other biomes. The Boreal cropland we analyzed, however, is concentrated in southern Canada (Figures S4 and S5). In terms of global representativeness of cropland, the 75% threshold also forces our analysis in the Temperate biome to central North America, mostly excluding extensive croplands in Europe, the Black Sea region, and northeastern China. The cropland in Grassland-Savanna, on the other hand, is well distributed across the continents even after the 75% threshold.

Cropland LULC is more affected than other LULC types by the 75% threshold. Table 2 shows how many grid cells (at the global scale) are included in our analysis for different threshold values, including the analysis threshold of 75%. Increasing the threshold from 50% to 75% results in over 86% more cropland grid cells being excluded from the analysis, whereas an average of less than 30% of grid cells are further excluded for other LULC types. Single grid cells with a high fraction of cropland are less common than grid cells with high fractions of other LULC types. Table S2 shows how the 50% and 75% thresholds affect the number of fire counts in our analysis.

**Table 2.** The number of grid cells in our analysis that meet various LULC thresholds. A threshold of 0% would be the entire dataset, whereas a threshold of 50% means that the grid cell is comprised of at least 50% of that LULC type. Our analysis threshold is 75%.

|  | Grid Cells Meeting 0% Threshold ($10^6$) | Grid Cells Meeting 50% Threshold ($10^6$) | Grid Cells Meeting 75% Threshold ($10^6$) | Fraction Remaining with 50% Threshold (%) | Fraction Remaining with 75% Threshold (%) |
|---|---|---|---|---|---|
| **CROPLAND** | 20.4 | 3.0 | 0.42 | 14.6% | 2.0% |
| **PASTURE** | 22.7 | 7.1 | 4.2 | 31.4% | 18.4% |
| **GRASS** | 19.8 | 9.2 | 6.9 | 46.3% | 34.7% |
| **FOREST** | 9.7 | 7.2 | 5.7 | 74.4% | 59.1% |
| **TOTAL** | 72.7 | 26.5 | 17.2 | 36.5% | 23.7% |

Figure 5 shows how relaxing the threshold requirement to 50% increases the global representativeness of crops considered in our analysis, which is perhaps more important than the number of grid cells being considered when building a general analysis of cropland fire–SM relationships. Comparing Figures 2c and 5 reveals that cropland with a relaxed threshold is more evenly distributed across the globe, and this is also true within each biome. With a 50% threshold, Boreal cropland is sampled from North America and southern Siberia, Temperate cropland from North America, Europe, the Black Sea region, and China, and Tropical cropland from Southeast Asia and West Africa. Other LULC types besides cropland sample relatively evenly from the unfiltered spatial distributions (Figure S4) for both 50% and 75% thresholds.

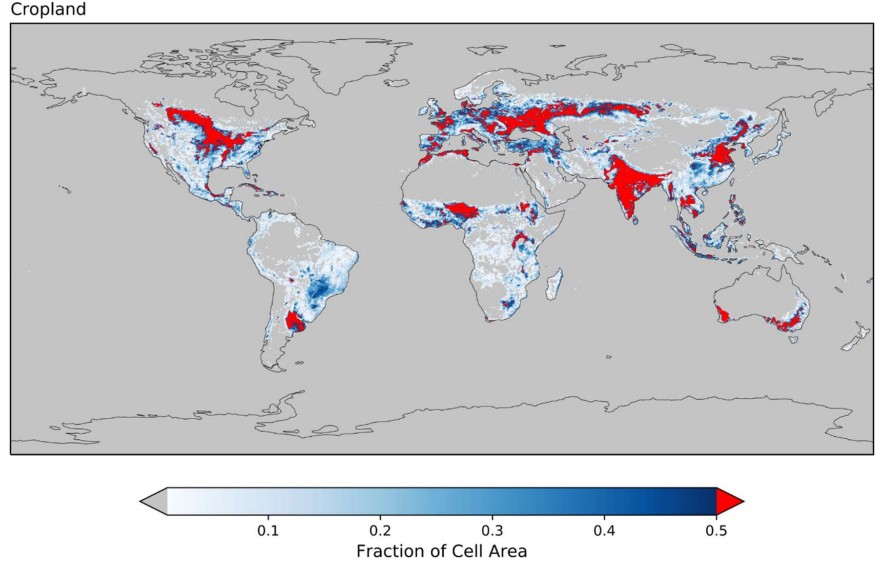

**Figure 5.** The global distribution of cropland land use and land cover. Red indicates grid cells with cropland fractions greater than 50%, and grey denotes that the grid cell has zero cropland. This is similar to Figure 2c, but with a lower threshold for a grid cell to be colored red.

We evaluated the quantitative effect of the cropland threshold on the fire–SM curves across the biomes to understand the sensitivity of the results in Figure 4c. Figure 6 shows the fire–SM relationship for cropland in each biome as a function of threshold values varying from 50% to 75%, where the 75% value corresponds to the results in Figure 4c. Recall that as the threshold decreases, the risk in data interpretation is that the FCs on that grid cell are less likely to be completely attributable to that LULC type. That being said, fires on cropland are perhaps the most heavily controlled by human presence, so we would speculate that it is likely that fires occurring in a crop-dominated landscape (a landscape with greater than 50% cropland) are likely to be management fires. We think this speculative reasoning supports a stricter threshold (i.e., a 75% threshold) for other LULC types, while, at the same time, allows for more flexibility (i.e., as low as 50%) on cropland LULC.

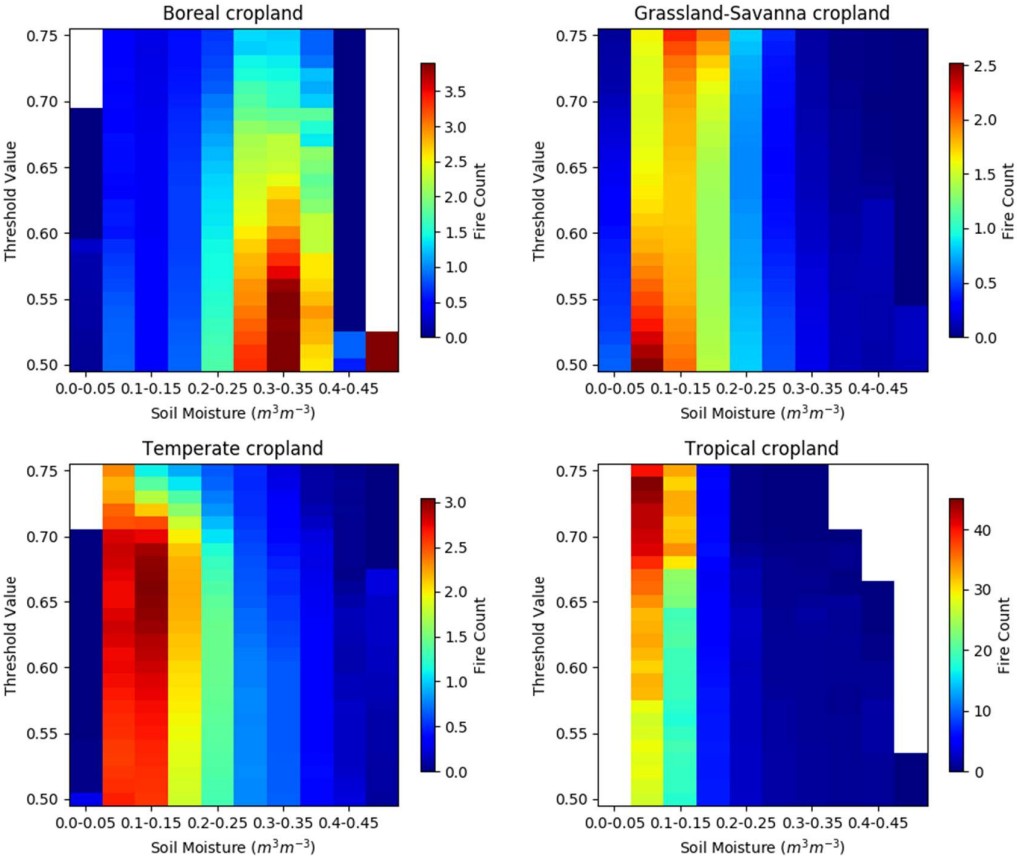

**Figure 6.** Heat maps showing the fire counts as a function of the threshold value and soil moisture for cropland in the four analysis biomes. Note that the fire count color scale varies.

The characteristics of the fire–SM curves in Figure 6 that are important are the location of the peak average FC in the SM dimension (shown with warmer colors) and the overall shape of the fire–SM curve itself as it departs from the peak to lower and higher SM values. Boreal fire–SM curves for cropland exhibit similar behavior across a wide range of thresholds, where the peak average FCs occur when SM ranges from 0.25 to 0.40 m$^3$/m$^3$, and the overall shape tends to flatten as the threshold decreases. The peak for cropland fire–SM curves in Grassland-Savanna migrates to lower SM values as the threshold decreases, but with the same overall shape. Temperate cropland fire–SM curves have a peak that migrates to slightly higher SM values as the threshold decreases, and develop a shape more similar to the cropland fire–SM curves for Grassland-Savanna, but extending to slightly higher SM values. Tropical cropland has a peak in the fire–SM curve that remains relatively stationary in the SM dimension as the threshold decreases, and the shape of the curve remains relatively similar even as the magnitude of FCs decreases.

Our analysis of fire–SM curves for croplands produces a broadly similar structure within a biome regardless of the threshold used for analysis, even though Figure 5 shows that there is a very different spatial distribution of the analyzed data when the threshold is relaxed. This lends some weight to the idea that our results are not strongly dependent on the analysis choices. For cropland in the Boreal biome, and perhaps in any biome, the fire–SM relationship could be investigated at sub-regional spatial scales e.g., [57] to evaluate how croppings [58] or documented burning practices with crop residue [59] affect the fire–SM curves in this LULC type. However, this kind of granular analysis is best left to a study that specifically considers regional and cultural practices.

*3.4. Pasture Land Cover*

About 84% of pasture LULC analyzed is in the Grassland-Savanna biome (similar to cropland), and the remaining 16% is distributed across the Boreal, Temperate, and Tropical biomes (Table 1). The spatial representativeness is less of a concern with pasture than it is with cropland. Boreal pasture is mostly densely located on the Tibetan plateau and south-central Siberia. Temperate pasture that we analyzed is scattered around the world in regions north of the Caspian Sea, Madagascar, small parts of southern and northern Africa, central South America, southern USA, northern Australia, and near southeastern Mongolia. Tropical pasture is concentrated in tropical southern Africa, South America, and southern Madagascar. Grassland-Savanna pastures analyzed in our study are distributed widely across all continents. Figure S4 shows these spatial distributions as a function of biome.

The global pasture fire–SM relationship is weighted heavily towards the fire–SM curve for the Grassland-Savanna biome (Figure 4d) where 84% of pasture being analyzed is located (Table 1), but disaggregating this global fire–SM curve into the more marginally contributing biomes reveals more structure. All pasture fire–SM curves show a distinct drop-off in average FC at low SM (less than 0.05 m$^3$/m$^3$) that is consistent with climate conditions that do not produce fuels, cultivated or otherwise. This pervasive feature is consistent with the fuel-constraint dimension of the fire–productivity model.

The climate constraint, however, is inconsistent on pasture LULC, as is evident in the varying behavior at higher SM following the peak FC at about 0.05–0.15 m$^3$/m$^3$. Average FC being elevated at higher SM is consistent with past research [4], and may be a quantitative signal related to the imposition of human fire use on a landscape, in this case, to manage pasture even when the SM conditions are not necessarily conducive [28,29,60]. The result also suggests that climate (with SM as the proxy) as a determinant for when fires occur on pasture has a less defined SM window than for fires occurring on grasses and in most of the forests (Figure 4a,b). We suggest that the weaker climate constraint on the pasture fire–SM relationship, mostly in the Grassland-Savanna biome (Figure S4), is consistent with human management fires that circumvent the fire season and burn the land cover prior to peak flammability conditions, a practice that is well documented in parts of Africa and Australia [60–63].

Future work could focus the methodologies presented in this study on pastures in the sub-regions of the Grassland-Savanna biome or across different types of managed pasture [49], but our results highlight a relatively general finding that pasture fires occur with a weaker climate constraint than grass fires. This weaker climate constraint is useful to quantify using our fire–SM approach since pasture fires account for a large fraction of global burned area [37], but are poorly captured in modeling [5] and difficult to separate from grass fires in land management strategies.

## 4. Conclusions

In this study, we attempted to understand whether the fire–productivity relationship [22,25,26] is evident when studying 15 years of remotely sensed monthly fire counts (FCs) and near-surface soil moisture (SM) at the global scale, arguing that SM itself should reflect both fuel and climate constraints on fire activity. To do this, we partitioned the globe into land-use and land-cover (LULC) categories of forest, grass, cropland, and pasture and then further segregated the parts of the world that have a largely dominant LULC type in a single grid cell. We then attributed any fire that occurs in that relatively homogeneous grid cell to fire that occurred on that LULC type. We probed the hypothesis

further by partitioning the globe into four broadly defined biomes (Boreal, Grassland-Savanna, Temperate, and Tropical) to study the dependence of fire–SM behavior on LULC across biomes.

Our results show that the productivity and climate constraints vary as a function of LULC type and across biomes in different ways. Forest and grassland LULC fire–SM curves follow behavior similar to the constraints on fire activity imposed by the balance of productivity and climate conditions, but forests generally have a narrower plateau for peak fire activity than grasses (Figure 4). Forests are more prone to fire only when there is a particularly low near-surface SM, most likely from moderate to extensive drought. In the Boreal biome, however, we believe that the shallower root systems of those tree species compared with trees in other biomes [39] result in a fire danger that is more directly connected to near-surface SM variability. Similar to boreal forests, grasses are more susceptible to fire over a wider range of SM values than forests across all biomes. Cropland fire–SM curves are not markedly different than those of grasses, with the exception of the crops in the Boreal biome. Fire danger on pastures extends to much higher SM values, which bypasses the climate constraint. We attribute this to human influence on the fire season on managed or partly managed pasture land cover.

Future work, in addition to the sub-regional analyses that incorporate detailed knowledge of pasture fraction or cropping, includes incorporating these observationally based fire–SM relationships into a modeling framework to test how well they capture the observed spatial and temporal distribution of fire. A stochastic dimension could be applied to account for the wide-ranging distribution of FC values (Figure S2) that extend from extreme fire behavior (hundreds of fire counts in a single grid cell) to a single fire occurrence. The stochastic dimension could impose a pseudo-natural variability in fires occurring on forests and grasses (consistent with the fire–productivity hypothesis) or on fires ignited, presumably intentionally, on pasture and cropland. In both cases, a stochastic approach would allow for the possibility of zero fire under high flammability conditions. A second component of future work could be to evaluate the role of antecedent soil moisture, similar to past studies that have been completed using biomes [22,64], on fire activity on all land-use and land-cover types.

Overall, our approach of using soil moisture and fire counts as a proxy for evaluating the fire–productivity relationship [22,26] for different land-use and land-cover types is a novel contribution that suggests an incremental approach to overcoming an important challenge in fire modeling [5,23,65] and paleoproxy data interpretation [7,8] by connecting human behavior with variability in the physical climate. Importantly, our results can be actively evaluated because both fire [46] and soil moisture [66] are directly observed at global scales using satellite sensors.

**Supplementary Materials:** The following are available online at http://www.mdpi.com/2571-6255/2/4/55/s1, Figure S1: Average number of points that were used to make the monthly average soil moisture of Figure 1 of the main text. Figure S2: Variability of the fire counts in each bin of soil moisture (first *y*-axis in raw fire counts ranging from zero to very high values), where the values on the second *y*-axis correspond to the colored lines in the figure and are also plotted in Figure 3 of the main text. Figure S3: Relationships between burned area and soil moisture across different land-use and land-cover types at the global scale. Figure S4: Fraction of grid cell for each LULC category (columns as Grassland, Forest, Pasture, and Cropland, as labeled) in each biome (rows, which are Boreal, Grassland-Savanna, Temperate, and Tropical biomes from top to bottom). Red indicates cells greater than the 75% threshold and grey denotes missing data. This expands on Figure 2 in the main text. Figure S5. Annual mean fire counts partitioned in the biomes used in this study, noting that the total mean annual fire counts are listed above each figure. Gray represents area outside of the biome, and white indicates that there are zero fire counts in that location. Figure S6: The relationships between fire counts and soil moisture across LULC types within each biome similar to Figure 4 in the main text but displayed per biome. Figure S7: Annual mean burned area partitioned in the biomes used in this study, noting that the total mean annual burned area is listed above each figure. Gray represents area outside of the biome, and white indicates that there is zero burned area in that location. Figure S8: The relationships between burned area and soil moisture across biomes for specific LULC types. Total represents the burned area–SM curve for the global data (in Figure S3, Supplementary Materials). Figure S9: The relationships between burned area and soil moisture across LULC types within each biome similar to Figure S8, Supplementary Materials, but displayed per biome. Table S1: Top table shows normalization constants for Figures 3 and 4 of the main text and Figure S6, Supplementary Materials, determined as the maximum bin-averaged FC for each LULC and each biome. The units are fire counts. The bottom table shows the corresponding normalization factors for the Figures S3, S8 and S9, Supplementary Materials, in units of $km^2$. Table S2: The percentage of fire counts (FCs) remaining as a function of LULC and biome after the

75% threshold is applied. In total (across biome and LULC), about 15% of FCs remain in the analysis with the 75% threshold. The numbers in brackets show the percentage of FCs remaining after a 50% threshold is applied.

**Author Contributions:** Conceptualization, A.J.S.; Data curation, A.J.S.; Formal analysis, A.J.S. and B.I.M.; Investigation, A.J.S. and B.I.M.; Methodology, A.J.S. and B.I.M.; Resources, B.I.M.; Software, A.J.S.; Visualization, A.J.S. and B.I.M.; Writing–original draft, A.J.S. and B.I.M.; Writing–review and editing, A.J.S. and B.I.M.

**Funding:** This research was partly supported by National Science Foundation Geography and Spatial Sciences BCS-1437074.

**Acknowledgments:** We thank the reviewers for their constructive comments on this version and a previous version of the manuscript.

**Conflicts of Interest:** The authors declare no conflict of interest.

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
