# Peer review of "Land-Cover Dependent Relationships between Fire and Soil Moisture"

_fire, doi:10.3390/fire2040055_

Round 1
Reviewer 1 Report
General Comments:
I found the paper interesting as it investigates the impacts of land cover and biomes on the fire-SM relationship. The authors divided the globe land cover into four major categories and found out that forest and grassland cover showed a similar fire-SM relationship. In the context of biomes, pasture fire was evident at higher soil moisture values compare to others.
Specific comments:
Introduction: Significant improvement required. The authors described different factors affecting fire occurrence. Please provide more information on the relationship between climate (precipitation, soil moisture, and temperature) and fire count. Please provide details on the present knowledge of land cover changes associated with fire and soil moisture and their limitations. The aim of the paper should be closely related to the gap of knowledge identified in the literature review.
Materials and methods: It is not clear to how the author counts the fire event; please provide the algorithm details for fire detecting using MODIS data. Please also provide more information about MODIS data collection and Figure 1.
Results and Discussion:
It was interesting to see the variation of fire-SM relationship with the land cover; however, I found the results were confusing and repetitive. It was hard to follow when the author pointed out the supplement figures in the main text, I would suggest including those in the main text. The finding of the paper depends on the monthly value of soil moisture; however, the distribution of the soil moisture varies before and after the fire event. I think author should use soil moisture anomalies instead of monthly soil moisture values for the analysis. It is not clear why a higher number of fire count, although high soil moisture value was noticed over the pastureland cover region. The threshold idea for the identifying different land cover seems to be confusing, could you please explain why it is required. In addition, the impact of the threshold should be discussed in the separate section rather than mentioned in the cropland biomes results section. As a secondary main suggestion, I would include a discussion section describing the potential implications and limitations of the study and mention how your results compare to another studies, which publish recently.

Author Response
Note that in addition to our replies below, we also replied to a number of relatively minor comments made directly in the PDF.
General Comments:
I found the paper interesting as it investigates the impacts of land cover and biomes on the fire-SM relationship. The authors divided the globe land cover into four major categories and found out that forest and grassland cover showed a similar fire-SM relationship. In the context of biomes, pasture fire was evident at higher soil moisture values compare to others.
Specific comments:
Introduction: Significant improvement required. The authors described different factors affecting fire occurrence. Please provide more information on the relationship between climate (precipitation, soil moisture, and temperature) and fire count. Please provide details on the present knowledge of land cover changes associated with fire and soil moisture and their limitations. The aim of the paper should be closely related to the gap of knowledge identified in the literature review.
In our introduction paragraph 2, we cited about 15-20 papers describing the literature about fire-climate interactions. In paragraph 3, we discussed the fire-productivity relationship, and in paragraph 4, we described connections of fire seasonality with inferred human influences on fire regime. Our study is the first to quantitatively connect soil moisture and fire as a function of land cover land-use type at large scales.
Materials and methods: It is not clear to how the author counts the fire event; please provide the algorithm details for fire detecting using MODIS data. Please also provide more information about MODIS data collection and Figure 1.
The fire count data products are available and completely described in a long trail of literature since MODIS extends back to the year 2000. We cited the most recent algorithmic advances, but we think our paper is more about what we can do with fire count data rather than how fire count data is collected. In response to your suggestion, we added text to this paragraph clarifying how MODIS uses thermal IR.
Results and Discussion:
It was interesting to see the variation of fire-SM relationship with the land cover; however, I found the results were confusing and repetitive. It was hard to follow when the author pointed out the supplement figures in the main text, I would suggest including those in the main text.
We thought very carefully about the inclusion of additional figures, but we think the paper is stronger with fewer figures in the main text. The supplement includes much more detail than seems needed for the main text.
The finding of the paper depends on the monthly value of soil moisture; however, the distribution of the soil moisture varies before and after the fire event. I think author should use soil moisture anomalies instead of monthly soil moisture values for the analysis.
We agree about before-after variations. Our first study of fire-SM is intended to provide a glimpse of this interconnectivity, and we will work to study higher time resolution data as well as antecedent soil moisture, both of which we suggest in the Conclusions.
It is not clear why a higher number of fire count, although high soil moisture value was noticed over the pastureland cover region.
We think this is an indication of the human influence over a managed land cover type, and cite examples of this sort of fire practice in the paper.
The threshold idea for the identifying different land cover seems to be confusing, could you please explain why it is required. In addition, the impact of the threshold should be discussed in the separate section rather than mentioned in the cropland biomes results section.
In paragraph 1 of Section 2.5, we stated that “since LULC is defined as a fraction of the grid cell, we only analyzed grid cells with LULC greater than a threshold value. In grid cells that met the threshold requirement, we assumed all the FC were associated with that particular LULC type. We tested LULC thresholds ranging from 50% to 100%, and selected 75% to balance two points: 1. Maximize the confidence that the FC in the grid cell are actually associated with that LULC type, and 2. Preserve enough data for meaningful results.”
As a secondary main suggestion, I would include a discussion section describing the potential implications and limitations of the study and mention how your results compare to another studies, which publish recently.
We discussed implication and limitations throughout Section 3 (Results and Discussion). We opted to discuss implications as a function of land cover type, but we also summarized our findings in the Conclusion section.

Reviewer 2 Report
This is a very interesting paper demonstrating the relationship between fire occurrence and remotely-sensed soil moisture globally. The authors calculated the distribution of fires across a range of soil moisture within separate biomes and land cover types. They also interpreted why these distributions look different in different places. These analyses and interpretations are helpful for understanding large-scale drivers of fire behavior.I believe this paper will be useful to many people in the fire management and fire modeling communities.
General Comments:
While the normalized fire counts are helpful for comparing distributions between biomes and land cover types, I would also like to have some information on the relative number of fires within each category. If some biomes only have a very small number of fires total, then the shapes of the distributions may not be as informative since each average could be skewed by one or two anomalous values. It would be helpful if the authors could include the total number of fires for each biome and land cover type somewhere in the paper.
Overall the information is presented well, but there are certain places where more information is needed to fully understand the authors' meaning. Please see the detailed comments below.
Detailed Comments:
Line 91: It is not necessary to say "We describe our methods ... discuss our results." I suggest deleting this sentence to improve the flow of the introduction.
Lines 104-108: This sentence is very long and difficult to follow. Please re-phrase. Also, it seems to me that the difficulty of retrieving soil moisture from densely vegetated areas would influence the area that could be mapped, but not the number of daily data points within each monthly mean. Please clarify the relationship between variable numbers of retrievals in different months and the vegetation density. In addition, it is not clear to me whether this monthly data set includes only 12 monthly means averaged across all years, or if there are separate maps for each month in each year (i.e., are there 12 maps of soil moisture total, or 12 maps for each year?). If there are only 12 maps, the authors should clarify why they chose to average across all years, since this would not allow them to capture the impacts of unusually dry or wet years.
Line 108: Please briefly explain what Forkel et al. mapped, for readers who are not familiar with that study.
Figure 1: I do not think it is necessary to show the soil moisture for every month, since it is difficult to see differences from month to month with the naked eye. Perhaps show only January, April, July, and October to capture one month from each season?
Lines 131-132: It is not clear why you bring up the Li et al. paper. Do you just mean to say that there are other data sets available? I think you should either explain what Li et al. found in their comparison, or just include Li et al. as a parenthetical citation at the end of a sentence stating that there are other land cover data sets that you could have used.
Lines 175-176: I suggest deleting the second half of this sentence, from "as would be expected" to "any SM value." This part of the sentence does not add any new information; it basically explains that the FC vary widely because the numbers are highly variable, which is a tautology.
Figure 3: I suggest removing the "1.2" label from the y axis, since the numbers are constrained to be no greater than 1. Also, the text that says "Global" at the top of the y axis is not necessary.
Line 193: Is "grass" at the end of this line supposed to be "pasture"? It is true that grass has higher fire danger for large SM values compared to forest, but not as high as pasture.
Line 194: Change "large values of SM values" to "large SM values."
Line 230: Do the authors mean "tend toward the global grass curve"? The phrase "tends towards the grass curve" is confusing since all of these curves represent grass.
Line 237: What is meant by "our biome"?
Line 286: I believe this sentence should say "After the 75% threshold is applied for the tropics, only 0.3..." or something to clarify that this is referring to the tropics.
Figure 6: I like this sensitivity analysis; it is very helpful for validating the methods.
Author Response
This is a very interesting paper demonstrating the relationship between fire occurrence and remotely-sensed soil moisture globally. The authors calculated the distribution of fires across a range of soil moisture within separate biomes and land cover types. They also interpreted why these distributions look different in different places. These analyses and interpretations are helpful for understanding large-scale drivers of fire behavior.I believe this paper will be useful to many people in the fire management and fire modeling communities.
General Comments:
While the normalized fire counts are helpful for comparing distributions between biomes and land cover types, I would also like to have some information on the relative number of fires within each category. If some biomes only have a very small number of fires total, then the shapes of the distributions may not be as informative since each average could be skewed by one or two anomalous values. It would be helpful if the authors could include the total number of fires for each biome and land cover type somewhere in the paper.
We included the normalization constants (the maximum of the bin averaged fire counts) in Supp Table 1. Each biome’s annual mean fire counts are included on Supp Fig 5 (241,000 in Boreal, 1.86 million in Grassland-Savanna, 1.04 million in Temperate, 845,000 in Tropical).
Overall the information is presented well, but there are certain places where more information is needed to fully understand the authors' meaning. Please see the detailed comments below.
Detailed Comments:
Line 91: It is not necessary to say "We describe our methods ... discuss our results." I suggest deleting this sentence to improve the flow of the introduction.
Agreed. Thanks.
Lines 104-108: This sentence is very long and difficult to follow. Please re-phrase. Also, it seems to me that the difficulty of retrieving soil moisture from densely vegetated areas would influence the area that could be mapped, but not the number of daily data points within each monthly mean. Please clarify the relationship between variable numbers of retrievals in different months and the vegetation density. In addition, it is not clear to me whether this monthly data set includes only 12 monthly means averaged across all years, or if there are separate maps for each month in each year (i.e., are there 12 maps of soil moisture total, or 12 maps for each year?). If there are only 12 maps, the authors should clarify why they chose to average across all years, since this would not allow them to capture the impacts of unusually dry or wet years.
Thanks for highlighting our confusing text. We have a full monthly timeseries of soil moisture maps so we can compare SM and fire counts on a month-by-month basis. Figure 1 was supposed to convey the mean monthly values as a glimpse of the full monthly timeseries. We modified the text to clarify this point.
Line 108: Please briefly explain what Forkel et al. mapped, for readers who are not familiar with that study.
From the Forkel paper: “As soil moisture cannot be accurately retrieved underneath dense (tropical) forests, estimates are not available in all regions, and thus the dataset has spatial gaps. We excluded such grid cells in the full analysis. Soil moisture time series were aggregated to monthly mean values. Temporal gaps in soil moisture time series were filled using a season-trend regression model as described in Forkel et al. (2013) and based on Verbesselt et al. (2010a, b), but without accounting for breakpoints. However, some years in some grid cells were excluded from the entire analysis if soil moisture estimates were only available for less than 3 months within this year. We used the monthly soil moisture values and long-term soil moisture conditions as predictor variables (Table 1). Long-term soil moisture conditions were computed as the mean soil moisture of the actual month and the 12 preceding months.”. We expanded on our summary of their results in the main text in Section 2.1. Thanks.
Figure 1: I do not think it is necessary to show the soil moisture for every month, since it is difficult to see differences from month to month with the naked eye. Perhaps show only January, April, July, and October to capture one month from each season?
We tried a version of the figure with only 4 months, like suggested, but this missed showing the subtle geographic shifts in the average climatology of SM when all 12 months are used. One way we will work with the journal to do is make the SM, FC, and LUH2 data that we used in our study available via a supplemental downloadable netcdf. This would allow anyone to delve into the details as needed.
Lines 131-132: It is not clear why you bring up the Li et al. paper. Do you just mean to say that there are other data sets available? I think you should either explain what Li et al. found in their comparison, or just include Li et al. as a parenthetical citation at the end of a sentence stating that there are other land cover data sets that you could have used.
Thanks, we agree that this was distracting. We had intended to simply acknowledge a key component of LUH2 (pasture) and that other land cover datasets exist.
Lines 175-176: I suggest deleting the second half of this sentence, from "as would be expected" to "any SM value." This part of the sentence does not add any new information; it basically explains that the FC vary widely because the numbers are highly variable, which is a tautology.
Thanks, that’s a fair point. Change made.
Figure 3: I suggest removing the "1.2" label from the y axis, since the numbers are constrained to be no greater than 1. Also, the text that says "Global" at the top of the y axis is not necessary.
Done. Thanks for pointing that out.
Line 193: Is "grass" at the end of this line supposed to be "pasture"? It is true that grass has higher fire danger for large SM values compared to forest, but not as high as pasture.
Yes, that was a mistake. Thanks for catching that.
Line 194: Change "large values of SM values" to "large SM values."
Thanks, done.
Line 230: Do the authors mean "tend toward the global grass curve"? The phrase "tends towards the grass curve" is confusing since all of these curves represent grass.
Yes, that was a grammatical mistake on our part. We revised the text.
Line 237: What is meant by "our biome"?
We revised the text to clarify the point we were trying to make that grass in the Temperate biome is largely isolated to a single location in DRC that itself is a borderline grass land-cover type.
Line 286: I believe this sentence should say "After the 75% threshold is applied for the tropics, only 0.3..." or something to clarify that this is referring to the tropics.
Thanks, we edited the text as suggested.
Figure 6: I like this sensitivity analysis; it is very helpful for validating the methods.
Thanks.
Round 2
Reviewer 1 Report
I am happy with the revised version of the paper, Thanks.